# Impacts of an Altimetric Wave Data Assimilation Scheme and Currents-Wave Coupling in an Operational Wave System: The New Copernicus Marine IBI Wave Forecast Service

Cristina Toledano [1,*], Malek Ghantous [2,3], Pablo Lorente [1,4], Alice Dalphinet [2], Lotfi Aouf [2] and Marcos G. Sotillo [4,5]

1   Nologin Consulting SL, NOW Systems, 50018 Zaragoza, Spain; pablo.lorente@nologin.es
2   Meteo France, 31057 Tolouse, France; mghantous@groupcls.com (M.G.); alice.dalphinet@meteo.fr (A.D.); lotfi.aouf@meteo.fr (L.A.)
3   Collect Localization Satellites, 31520 Ramonville-Saint-Agne, France
4   Puertos del Estado, Área de Medio Físico, 28042 Madrid, Spain; marcos@puertos.es
5   Mercator Ocean International, 31400 Tolouse, France
*   Correspondence: cristina.toledano@nologin.es

**Abstract:** The Copernicus Marine IBI-MFC (Iberia–Biscay–Ireland Monitoring and Forecasting Centre) has delivered operational wave forecasts since 2017. The operational application is based on a MFWAM model (Meteo-France WAve Model) set-up, running at a $1/20^\circ$ grid (5-km). The research presented here was conducted to improve the accuracy of the IBI-MFC wave model products, by means of (i) including a new wave data assimilation scheme and (ii) developing a new coupled ocean-wave modelling framework. Evaluation of these set-up upgrades, in terms of improvements in IBI wave model system capabilities, is here presented. All the model sensitivity test runs, performed for the year 2018, are assessed over the whole IBI domain, using the available in-situ (from 49 mooring buoys) and independent satellite wave observation. The results show that the most relevant improvement is due to the data assimilation, while the impact of surface ocean currents, although less significant, also improves the wave model qualification over the IBI area. The demonstrated benefit, related to the herein proposed upgrades, supported the IBI-MFC decision to evolve its operational wave system, using (since the March 2020 Copernicus Marine Release) the resulting wave model set-up, with data assimilation and currents-wave coupling for operational purposes.

**Keywords:** forecasting; wave modeling; data assimilation; current–wave coupling; current forcing; model validation; wave altimetric products; Copernicus Marine IBI

## 1. Introduction

Waves constitute the interface between ocean and atmosphere and have an important role in terms of exchanges through this interface [1]. Their representation is necessary to accurately compute the different air–sea fluxes of heat and momentum [2].

There is a widespread worldwide offer of accurate and reliable wave forecast services. A variety of operational wave forecast services, ranging from global to local coastal scales, are run by different operational oceanographic centres (some of them national weather offices); the wave forecast products benefit different end-users, supporting day-to-day operations at sea and contributing to warning systems that minimize potential risks for marine safety (among others). The authors in [3], in their review of European Operational oceanographic capacities, indicated how several wave models (i.e., WAM (Wave Model), SWaN (Simulation Waves Nearshore), WaveWatch-iii, WWM-II, etc.) [4–6] are used in the forecast services delivered by the operational oceanographic centres. Some of these operational services use operational assimilation schemes to account for near real-time observational wave information, especially from satellite altimeters [7].

Upper ocean dynamics are strongly affected by sea-state dependent processes, inducing the impact of waves on the ocean's small- and large-scale circulation.

On the one hand, waves affect the ocean surface layer through different processes [8]: Waves induce surface current via the Stokes drift, adding a term on the Coriolis effect in the momentum equation (the so-called Stokes–Coriolis force). Part of the atmospheric wind stress contributes to the wind-wave growth, thus, subtracting a quantity of energy to the ocean currents. Furthermore, during wave breaking, turbulent kinetic energy is produced and affects the upper ocean surface layer, enhancing the turbulent mixing. Recent studies have attempted to determine the impacts of wave effects on the representation of the ocean surface layer at different spatio-temporal scales. Among others: wave-induced mix-layer depths representation [9], relevant impacts on the atmospheric surface temperature, pressure, and precipitation [10,11], modifications in wind stress by the rise of roughness length and friction velocity [12]. This is especially true during storm events, when wave–current interactions might represent a leading order process of the upper ocean. In this context, ref. [13] strongly recommends using an ocean-waves-atmosphere coupled system to improve the representation of tropical cyclones' intensity, structure and motion. Indeed, ref. [14] studied the effect of sea waves on the typhoon Imodu (15–19 July 2003). Moreover, ref. [15] demonstrated how a coupled system simulates more accurate surface dynamics than uncoupled models, with larger improvement on the shelf, showing that (especially during extreme events) ocean-wave coupling improves the accuracy of the surface dynamics, with larger improvements in the simulation of ocean currents over the shelf due to the synergy between strong tidal currents and more mature decaying waves.

On the other hand, the presence of ocean currents affects the waves, changing their amplitude, frequency and direction. This is generally due to the energy bunching, accounted in the wave energy balance when the velocity of the wave energy propagates across the current, the energy transfer between waves and currents, the frequency shifting (including Doppler shifting) and current-induced refraction [16]. Ref. [17] accounts for significant wave height changes in the Baltic Sea due to the impact of ocean currents (up to 20% in specific severe storm conditions, mostly in shallower waters and when waves and surface currents propagate in opposite directions [18,19]). The Copernicus Marine Service [20,21], one of the streamlined six thematic streams of the Copernicus Services (Atmosphere, Marine, Land, Climate Change, Security and Emergency) [22,23] and internationally recognized as one of the most advanced service capabilities in terms of ocean monitoring and forecasting, provides regular systematic reference information on the physical, biogeochemical and sea-ice state for the European regional seas and the global ocean. This service recently included, in its product portfolio, essential ocean variables related to the sea state, and near-real-time wave forecasts, and multi-year wave reanalysis products were progressively incorporated (along the 2015–2018 development phase) in the Copernicus Marine Service offer. The Copernicus Marine Service high-level strategy includes a roadmap with associated Research and Development (R&D) priorities [24], which identifies some developments per thematic area that are key for the future service evolution. Among others, (i) upgrade of data assimilation schemes (to improve the analysis and reanalysis capabilities) and (ii) enhancement of the representations of coupling effects between ocean-wave-sea-ice-atmosphere-land components (to improve forecast model solutions) are seen as prioritized research lines for any Copernicus Marine Monitoring and Forecasting Centres (MFC).

Specifically, for the European Atlantic Façade, the Copernicus Marine IBI-MFC (Iberia–Biscay–Ireland Monitoring Forecasting Centre) delivers daily ocean model estimates and forecasts of different physical and biogeochemical parameters, including, since 2016, hourly wave forecasts and multi-year products [25].

The present work focuses on the research performed to develop the current operational version of the IBI-MFC wave model application. This research was mainly conducted to improve the accuracy of these IBI-MFC wave model products, by means of developing a

new coupled ocean-wave modelling framework that also includes wave data assimilation. In that sense, this study has two specific objectives:

- To assess and quantify the potential added value, in terms of accuracy gain, that the assimilation of altimetric significant wave height satellite observation has on the IBI wave model solution.
- To analyse the impacts in the IBI wave model solution related to the use of surface current–wave coupling, evaluating the contribution of surface ocean currents in the wave energy balance.

To address these questions, different wave model sensitivity tests are performed. Several wave model simulations generated with the IBI-MFC wave model set-up, and only differing from the operational version (available in 2018) for the activation of the new data assimilation scheme and in the degree of the ocean current forcing applied, are run. The assessment of these model simulations is conducted using several local available in-situ and satellite wave observations.

The paper is organised as follows: Section 2 provides a description of the Copernicus IBI-MFC wave model system and the different model sensitivity tests performed, together with the model assessment proposed. Section 3 presents the main results, with an analysis of the proposed updates. Finally, in Section 4, the impacts of the current forcing interactions and the data assimilation scheme proposed are discussed, providing a look ahead to related benefits on the IBI wave operational forecast capabilities.

## 2. Methodology and Sensitivity Tests for Copernicus Marine IBI-MFC Wave System

### 2.1. The IBI Area and IBI-MFC Wave Model

The Copernicus Marine IBI-MFC (Iberia–Biscay–Ireland Monitoring Forecasting Centre) offers a comprehensive portfolio of regular and systematic regional information on the state of the ocean for the European Atlantic façade, supporting all kinds of marine applications. As part of this IBI-MFC service, a short term (10-days) high-resolution wave forecast is updated twice a day for the IBI area. Hourly instantaneous data for significant wave height, wave direction, wave period variables, together with wind sea and swell (primary and secondary partitioned wave spectra) parameters are delivered as part of this regional Copernicus Marine IBI wave product.

The MFWAM model configuration for IBI MFC is implemented on the IBI domain (26–56° N and −19–5° E; see geographical domain in Figure 1) with a horizontal resolution of 5 km approximately (1/20°).

The wave model used as base of this IBI-MFC operational system is the MFWAM [26]. This MFWAM model is based on the IFS-ECWAM 41R2 code [27], with changes regarding the dissipation by wave breaking and the swell damping source terms as developed by [28]. The current version of the model includes major improvements achieved within the FP7 European research MyWave Project [2,29]. The IBI-wave model performs a partitioning technique on wave spectra over all ocean grid points of the IBI domain. The partitioning technique is based on the watershed method developed for image processing [30]. This process effectively treats the wave spectrum as a topographic map from which individual peaks in wave energy can be identified to define the separate wave components. First, wave spectrum is split in wind sea and swell wave spectra. Then, partitioning is applied for the swell wave spectrum. The peaks on the spectrum are isolated and they are considered as partitions. Afterward, classification of swell partitions in primary and secondary swell is performed depending on the mean energy of each partition.

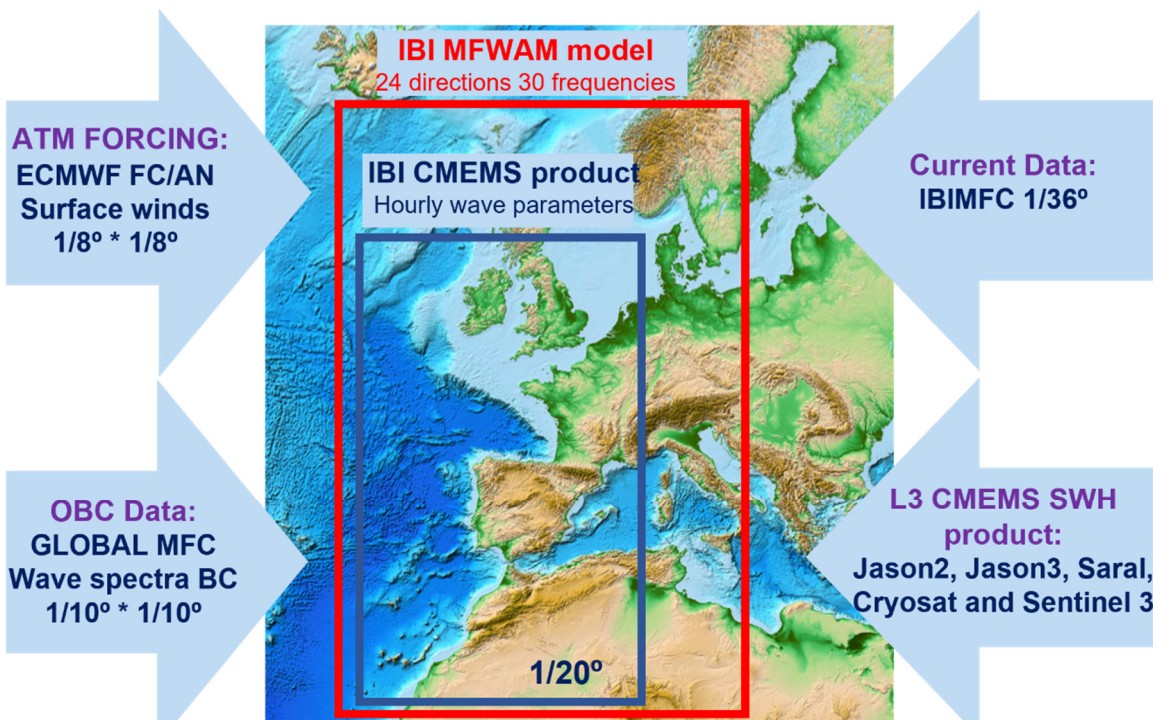

**Figure 1.** The CMEMS IBI-MFC wave Forecast/Analysis model Application. Model and product service spatial coverages, and details on the forcing, Open Boundary conditions and external data sources used for the ocean current forcing and the Data Assimilation applied.

The IBI-MFC wave model was upgraded (January 2018 Operational release) to improve the drag coefficient variation with the wind speed, resulting in positive impacts in the surface stress characterization. This improvement of the surface stress is certainly needed for the coupling with the IBI ocean currents here tested. To this end, a new setting on the wave dissipation term, the sheltering parameter, and the use of Phillips spectrum tail for the high-frequency part of the wave spectrum was also implemented. Moreover, the minimum water depth is taken as 5 m (instead of the 1 m value used in former IBI wave model versions). Associated to these upgrades, slight improvements in terms of significant wave height were obtained (reduced scatter index, around 1.9% when comparing model results with observations from altimeters [31]).

The bathymetry used is derived from the 1 arc minute ETOPO1 ocean bathymetry by National Geophysical Data (NOAA) [32]. The wave spectrum is discretized in 24 directions and 30 frequencies ranging from 0.035 Hz to 0.56 Hz. The Copernicus Marine IBI wave model is driven by short-range forecasted and analyzed IFS-ECMWF winds [33] at 1/8° hourly winds, which are used for the first 90 h, decreasing time frequency to 3-hourly until T + 144 h, and 6-hourly forecasts until T + 240 h. It uses as boundary conditions (wave spectra) Copernicus Marine GLOBAL wave data at 1/10° spatial resolution [34].

An IBI-MFC wave forecast product (10 days hourly data updated twice a day) is delivered to end-users for the IBI service domain (see spatial coverage in Figure 1), together with a 2-year historic timeseries, composed of IBI best estimates (analysis data for the Day-1 date).

*2.2. Model Sensitivity Test: The Proposed IBI Wave Model Upgrades*

The Copernicus Marine IBI-MFC identified, among others, the following two major shortcomings in its wave forecast service:

(i)     The lack of IBI wave analyses to initialize IBI forecast cycles
(ii)    The absence of any coupling of the IBI wave model data with ocean currents.

Bridging these 2 identified gaps in the Copernicus IBI wave service was considered as a major goal for the IBI-MFC service evolution planned for the last Copernicus-1 Service Phase (2018–2021) and 2 different research working lines were followed to achieve the objective. These specific research lines were fully aligned with the Copernicus Marine Evolution Strategy [20], and their scientific research priorities (implementation of data assimilation schemes and enhancement of the model coupling between different earth system components) were two of the major amelioration axes proposed for the Copernicus Marine products and services.

The present work aims to quantify the potential added value of new IBI wave analyses (to be generated by means of a new data assimilation scheme implemented to assimilate altimetric significant wave height satellite observations) with respect to the IBI best model wave estimates (note that traditionally, the IBI-MFC was delivered as historic best model estimate 2 years of model hindcast wave data, wave run for day D–1 forced with analyzed winds). Likewise, the impact of including the contribution of the surface ocean currents on the IBI wave model solution is evaluated.

The impacts on the IBI wave solution of both the new IBI wave DA system and the current–wave coupling are evaluated through specific IBI-like wave model scenarios. To this aim, different model sensitivity test runs, based on the IBI-MFC operational model configuration, have been designed and run. Table 1 shows an overview of the four proposed IBI wave model scenarios: the Control run (IBI-CO) was performed using the same wave model set-up that was used in the IBI operations in December 2019. Two more IBI wave sensitivity test runs were performed: one with the current forcing activated, but without Data Assimilation (the IBI-CU run) and another with data assimilation activated, but without any coupling contribution (the IBI-DA run). Finally, a last model scenario that includes the two novelties (data assimilation and ocean current forcing) was assessed. This model scenario (hereafter named as IBI-OP) is consistent with the new IBI-MFC operational CMEMS IBI wave model system (in operations since July 2020).

**Table 1.** IBI wave model scenarios: overview of model runs performed to test the 2 proposed novelties (the current forcing and the new Data Assimilation scheme). The Control run and all the test runs using the IBI wave model set-up used in operations in Copernicus release in December 2019. The last sensitivity test, switching both the DA and the currents coupling, to be proposed as base of the new Copernicus IBI wave forecast system (in operations since July 2020).

| IBI-Wave Model Scenarios | Currents Coupling | Data Assimilation |
|---|:---:|:---:|
| IBI CONTROL RUN (IBI-CO) | - | - |
| IBI Run with currents forcing on (IBI-CU) | x | - |
| IBI Run with Data Assimilation on (IBI-DA) | - | x |
| New IBI Operational Wave system (IBI-OP) | x | x |

The four different model scenarios proposed were run over the year 2018. The IBI wave model set-up novelties were assessed, and all the proposed sensitivity model runs were validated using in-situ and remote sensing observations (i.e., from coastal and deep-water mooring buoys, and altimetric SWH). A complete description of the different sensitivity model tests performed to improve the operational wave model application that comprise the IBI-MFC operational wave forecast service is provided below. Results from the different model test runs are provided in the following section.

### 2.2.1. The Altimetric Wave DA Scheme Proposed for IBI

The data assimilation scheme proposed to be applied in the IBI wave service and tested through the IBI-DA test run presented here, is based on the optimal interpolation scheme described by [35] and it is the same scheme used in the Copernicus Marine GLOBAL wave system. The variable to be assimilated is the significant wave height, Hs. Because Hs is not a state variable of the system, this introduces an extra complication in that the energy

must be repartitioned from a frequency and direction integrated parameter (the Hs) to the full directional frequency spectrum. This involves making several assumptions and is by nature inexact, but in practice performs well [7]. What follows is a brief description of the method, which has been adjusted for the ST4 physics used in the IBI-wave model.

For a state vector x, optimal interpolation seeks a weighted combination of the background (or mode forecast), denoted by $x^f$, with observations, $y^o$, in order to produce an analysis $x^a$. The fundamental equation is:

$$x^a = x^f + \mathbf{K}\left(y^o - \mathbf{H}x^f\right), \tag{1}$$

where $\mathbf{H}$ is the observation matrix. $\mathbf{K}$ is a weighting matrix.

$$\mathbf{K} = \mathbf{P}\mathbf{H}^{\mathrm{T}}\left(\mathbf{H}\mathbf{P}\mathbf{H}^{\mathrm{T}} + \mathbf{R}\right)^{-1}, \tag{2}$$

where the matrices $\mathbf{P}$ and $\mathbf{R}$ are respectively the model and observation error covariance matrices.

In MFWAM these matrices are expressed as correlation matrices:

$$\mathbf{K} = \mathbf{C}\mathbf{H}^{\mathrm{T}}\left(\mathbf{H}\mathbf{C}\mathbf{H}^{\mathrm{T}} + \mathbf{I}\right)^{-1}, \tag{3}$$

The ratio of background and observation errors is kept constant over the IBI domain and set to 1 (i.e., model and observation errors are assumed to be equal everywhere). Although a different ratio may be theoretically justifiable, we have found that this value works best for this model in this domain; it is the same as that used in the global configuration. Observational errors are additionally assumed to be uncorrelated. With these assumptions, R is none other than the identity matrix, I. P becomes the correlation matrix $C$ defined in terms of the correlation length $\lambda_c$:

$$c_{ij} = e^{-\left(\frac{d_{ij}}{\lambda_c}\right)^a}, \tag{4}$$

where $d_{ij}$ is the great circle distance between points $i$ and $j$ and a is a tuning parameter.

With this simplified OI scheme the only tunable parameters are in the correlation function. For the IBI configuration used here the correlation length $\lambda_c$ is set to 170 km, significantly less than the 300 km length used in the global configuration, and the tuning parameter a is set to 1. We performed some experiments with $\lambda_c$ and $a$, in particular testing values determined from a correlation study of the global model divided by basin [36]. These did not result in an improvement in performance, however, so the original values were kept. The cutoff distance, beyond which observations are not included in the analysis, is set to 650 km. The analysis, Equation (1), produces a corrected estimate for the significant wave height. Since *Hs* is not a state vector of the wave model, but rather an integrated parameter, in order to correct the model itself MFWAM redistributes the energy in the wave spectrum using the method of [35], which is based on empirical wave growth laws. The analyzed spectrum is expressed as:

$$F^a = ABF^f(Bf, \theta), \tag{5}$$

where $F$ denotes the wave spectrum, $f$ the frequency and $\theta$ the direction, and the superscripts $a$ and $f$ refer to analysis and background respectively. $A$ is the ratio of analysis to background energy, which can be expressed as $(H_s^a/H_s^f)^2$, where $A$ determines the overall energy correction to the spectrum, while the effect of $B$ is to rescale the frequencies.

Two different methods for computing $B$ are used, depending on whether the spectrum is determined to be primarily a swell spectrum (the energy of the swell accounts for more than 1/4 of the total energy of the spectrum) or a wind–sea spectrum. If the spectrum is predominantly wind–sea, $B$ is computed from the mean frequencies of the background and analysis as:

$$B = \overline{f}^f / \overline{f}^a, \tag{6}$$

The choice of mean frequency was for computation reasons; the peak frequency would be just as valid a choice (if not more so), as both are in any case approximations. If the spectrum is predominantly swell, the average steepness of the waves is assumed constant. Therefore:

$$B = \overline{f}^f / \overline{f}^a = \sqrt{H_s^a / H_s^f},$$ (7)

The calculation of $\overline{f}^f$ and $H_s^f$ is taken care of by the model, but fa and $H_s^a$ have to be estimated from the *Hs* analysis. This is done by exploiting the empirically derived duration-limited growth laws. By defining the non-dimensional energy, mean frequency and duration, respectively:

$$\epsilon' = u_*^4 \epsilon / g^2,$$ (8)

$$f' = u_* f / g,$$ (9)

$$t' = tg / u_*,$$ (10)

where *g* is gravitational acceleration, and $\epsilon$ = Hs2/4. The growth relations are:

$$\epsilon'(t) = 1877 \left( t' \left[ t' + 5.440 * 10^5 \right] \right)^{1.9},$$ (11)

$$\epsilon\left(\overline{f}\right) = 5.054 * 10^{-4} \overline{f}^{-2.959},$$ (12)

From the background friction velocity and *Hs* we can use Equation (9) and the growth law equations to estimate an updated $\epsilon'$ and $f'$. These in turn give us $\overline{f}^a$ and $H_s^a$, and with these we can calculate *B* and *A* and produce the updated spectrum (5).

The data are combined to generate super-observations. Within a set time window around the analysis time, the data are assigned to model grid points closest to them, and any data sharing a model point are averaged together and treated as one observation. This reduces the number of data to assimilate, lightening the computational load, and it helps to smooth out potential errors in the observations. Outlying data are rejected in this step.

This data assimilation method results in a final *Hs* corrected mostly in the wind sea part of the frequency spectrum [7].

### 2.2.2. Wave–Current Coupling Proposed for IBI

To incorporate surface ocean currents forcing in the IBI wave model system, surface current data from the CMEMS-IBI analysis and forecast ocean model system were used as inputs in the IBI wave model system for wave refraction.

The presence of current may change the amplitude, frequency, and direction of waves. This is generally due to the energy bunching that is readily accounted for the energy balance equation of waves using the velocity of the wave energy propagating across the current, the energy transfer between waves and currents, the frequency shifting (including Doppler shifting) and current-induced refraction [16].

MFWAM model equations include the depth and current refraction. The propagation velocity in the relative frequency space should be computed according to [37]:

$$c_\sigma = \frac{\partial \sigma}{\partial d} \left[ \frac{\partial d}{\partial t} + \vec{u} \cdot \vec{\nabla} d \right] - c_g \vec{k} \cdot \frac{\partial \vec{u}}{\partial t},$$ (13)

where $\sigma$ is the relative frequency, $c_\sigma$ the propagation velocity in the relative frequency space, *t* is time coordinate, *s* is the space coordinate in the direction of propagation, $\vec{\nabla}$ the gradient operator in the geographical space, *d* is the total depth, $\vec{u}$ is the current, $c_g$ is the group velocity and $\vec{k}$ is the wave number vector. As in MFWAM model current and water depth are time-independent, the term $\partial d / \partial t$ in Equation (13) is not present.

The offline method of coupling with surface currents takes the files needed for the whole forecast period from the IBI-MFC ocean forecast system. The files used are not exactly

the ones delivered through the Copernicus Marine catalogue, but rather the native IBI NEMO model outputs (at the 1/36° ORCA grid), which the IBI wave system interpolates from the IBI ocean model grid into the 1/20° regular grid used for the MFWAM model.

2.2.3. Assessment of Model Runs: Evaluation Criteria against In-Situ and Altimeter Data

To assess the performance of the numerical model applying both novelties and to identify the main sources of uncertainty linked to the Wave–Current coupling and the application of the data assimilation in the IBI wave model simulations, the four different model scenarios were performed over the year 2018. The significant wave height and mean period fields, resulting from the different IBI sensitivity runs, are validated by means of comparison with different in-situ and satellite remote sensed observational data sources.

In-situ measurements of significant wave height, $H_s$, and mean wave period, $Tm_{02}$, were extracted from mooring buoys available in the IBI region, compiled in the product delivered by the Copernicus Marine IBI INSITU-TAC [38]. The mean wave period ($Tm_{02}$) is defined as follows:

$$Tm_{02} = 2\pi \left( \frac{\iint \omega^2 E(\omega,\theta)d\omega d\theta}{\iint E(\omega,\theta)d\omega d\theta} \right)^{\frac{-1}{2}},$$

(14)

where $E(\omega,\theta)$ is the variance density and w the absolute radian frequency.

Measurements for $Hs$ and $Tm_{02}$ variables are available for the examined year in the area at 49 and 45 buoys, respectively (see list and locations in Figure 2).

To validate the model through model output and buoy data collocation, the time series were taken at the model grid point nearest to the buoy location. For regional validation purposes, the IBI domain is split into different sub-regions of interest, being validation metrics gathered for the whole IBI service domain and for each sub-region (see spatial domains in Figure 2). Likewise, different metrics are computed separately using coastal and deep-water mooring buoys.

The different model sensitivity test runs were validated with satellite observations of significant wave height, Hs. However, since data from Jason-2, Jason-3, Saral, Cryosat-2 and Sentinel3 altimeters are now assimilated into the model (information from this mission included in the L3 CMEMS altimeter data product used for assimilation), an independent data source is needed. Thus, the diagnostic after data assimilation is performed by comparing the model to the HY-2A satellite altimeter processed by the French Space Agency CNES. The validation procedure with altimetric observations begins with pre-processing the $Hs$ data, rejecting $Hs$ data lower than 0.5 m or higher than 12 m and eliminating big jumps in terms of $Hs$ value from one observation to the next one (the biggest value of steps higher than 1 m in case of positive steps and −2 m, in case of negative steps, is rejected). To validate the model runs with altimetric observations we used the HY-2A satellite data for both DA and reference runs, ensuring that a data source independent from the assimilated data, HY-2A $Hs$ is biased [39], so a calibration to reference mission such as Jason-3 has been implemented on crossover locations. This leads to a linear for HY-2A Hs, which is expressed as follows:

$$H_s = 0.9476 \times H_s^{biased} - 0.0230,$$

(15)

The modelled $Hs$ is also post-processed with an upscaling to a 0.1 degree resolution (the nearest grid point for altimeters), in order to closer match the observations, and it is limited to values above 0.5 m. The final step to prepare for validation is then to find the points of modelled $Hs$ that correspond to the observation points. The validation statistics are then computed using these two values for each point where a valid observation exists.

Apart from bias, root mean square difference (RMSD) and correlation (CCOR), validation against altimeters includes the statistical quantity scatter index (*SI2*) used for the wave model statistics defined as:

$$SI2 = \frac{\sqrt{\sum_{i=1}^{N}[(x_i - \bar{x}) - (y_i - \bar{y})]^2}}{\sum_{i=1}^{N} y_i},$$

(16)

where $y_i$ is the observation, $x_i$ the model value corresponding to the $i$th observation, $N$ is the total number of observations, and the overbars refer to the population mean. This definition of scatter index differs from others in that the observations are not squared before taking the mean, so is only valid for quantities such as *Hs* which are always positive.

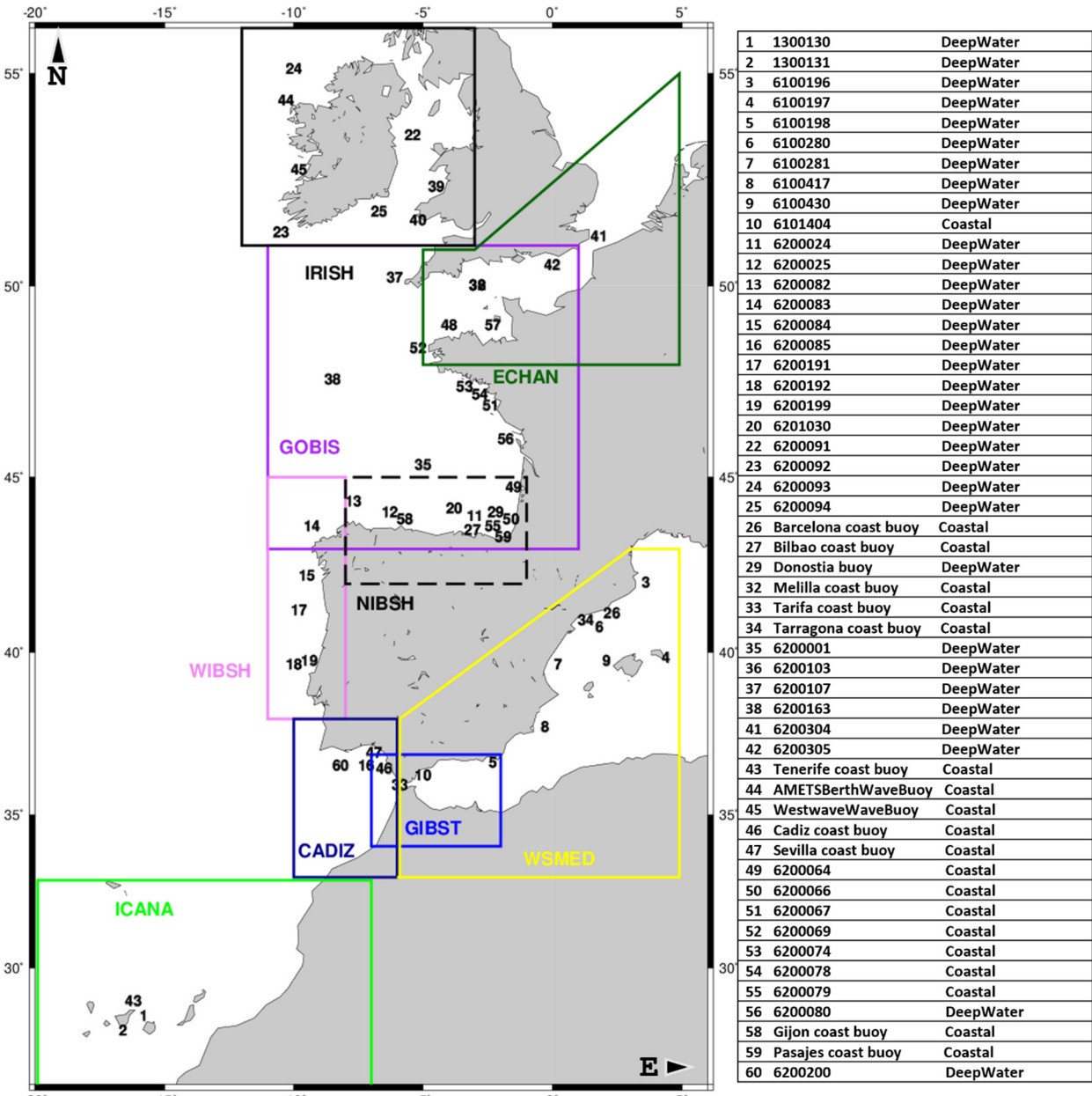

**Figure 2.** Locations of all the mooring buoys used in the model validation performed for the year 2018. In the accompanying table, the seven-digit WMO identifier (or buoy name provided by CMEMS) is followed by the information on the location of the mooring buoy (coastal or deep water). The 8 different sub-regions of interest for validation purposes in which the IBI service domain is divided are shown in the map (see polygons of different colors). From the list of 60 buoys in the IBI region, only the 49 available for year 2018 have been used.

A lower scatter index is not always a reliable gauge for model performance [40]. The symmetrically normalized root square error (*HH*), also used for validation against altimeters, provides more accurate information on the accuracy of simulation.

The error indicator *HH* proposed by Hanna and Henold, ref. [41], is defined as the RMSE divided by the absolute value of the mean of the product of the observations and modelled values:

$$HH = \sqrt{\frac{\sum_{i=1}^{N}(x_i - y_i)^2}{\sum_{i=1}^{N} y_i x_i}},$$ (17)

Then, in the next section, the impact for the two proposed novelties (current forcing and data assimilation) is assessed with tests performed in 2018, following the evaluation criteria against in-situ and altimeter data described above.

## 3. Results

### 3.1. Impacts in the IBI Wave Solution Related to the Use of Surface Current–Wave Coupling

For the assessment of the surface current–wave coupling, model outputs from the control run (IBI-CO; run with no currents coupling and no data assimilation) and the no data assimilation run with current–wave coupling (IBI-CU) were compared with in-situ observations, corresponding to the year 2018, focusing the model validation on the wave parameters of significant wave height, *Hs*, and mean wave period, $Tm_{02}$.

The error metrics from both model configuration test runs are summarized in Tables 2 and 3. Table 2 shows results of a comparison between the wave hindcast best estimate and in-situ observations, and Table 3 shows a similar comparison with altimeter data for the *Hs* parameter, with skill metrics differentiated between CMEMS L3 altimeters and HY-2A. The general trend of validation for Tables 2 and 3 is very similar due to the good correlation ($\geq$0.98) between buoys measurements and altimeters, as the comparison performed by [42] reveals.

The results shown in both tables do not show substantial improvement between the coupled and uncoupled model, but differences are definitely evident in terms of mean period. Coupling effects are stronger for coastal buoys locations, in some cases with worse metrics due to overestimation in the peaks of $Tm_{02}$ time series. On the other hand, the systematic negative bias in *Hs*, caused by the tendency of the IBI wave system to underpredict, is reduced with the current surface implementation.

Metrics in Strait of Gibraltar (CADIZ an GIBST subregions) have different solutions for coastal and deep-water buoys. In general, results are slightly better for the coupled model IBI-CU. This is due to current coupling increasing values of wave height, but in some coastal locations, when the $H_s$ is slight, an unrealistic ocean current overestimates the $Tm_{02}$.

Figure 3 depicts the pattern of agreement between the IBI wave system with wave–current coupling (IBI-CU) and no coupling (IBI-CO). As Table 2 shows, the western part of the IBI region (WIBSH, GOBIS and IRISH subregions) has analogous solutions: slightly better results for significant wave height and better metrics for the mean period, in the case of the coupled model (IBI-CU). At buoy 6200083, located on the western coast of Galicia (depth of 386 m), the comparison of model results serves as an explicit example of the general IBI performance. The effect of the current coupling is not very sensitive in the Hs, but bias for the $Tm_{02}$ drops from 0.53 to 0.46 and the RMSD from 0.72 to 0.60 (note the time series agreement except for the mean period on 9 December 2018). Similar error reductions are found throughout the year.

**Table 2.** Error metrics for the test runs with and without current–wave coupling (IBI-CU (CU) and IBI-CO (CO) test runs, respectively) computed with hourly observations at mooring buoys. Time period: 2018 Variables: Significant wave height (SWH) and mean wave period (TM02). Metrics computed for the whole IBI service domain and for the 9 Validation regions (i.e., IRISH, ECHAN, GOBIS, NIBSH, WIBSH, GIBST, CADIZ, WSMED, ICANA) used by the Copernicus Marine IBI-MFC service. Metrics are gathered using all the available buoys in each region, and also using exclusively Coastal and Deep-water Buoys (CB and DB, respectively). Each error metric (Bias, Root-Mean-Square differences (RMSD) Correlation (CCOR)) provided for each model solution. N counts the size of the sample. Bold numbers highlight the best performing dataset. Mooring buoys in the English Channel area give the zero-crossing wave period (Tz) instead of $Tm_{02}$, so mean period measurements for the ECHAN region are not provided for this validation.

| | | | SWH | | | | | | TM02 | | | | | |
| | | | BIAS (m) | | RMSD (m) | | CCOR | | BIAS (s) | | RMSD (s) | | CCOR | |
| | | N | CU | CO | CU | CO | CU | CO | CU | CO | CU | CO | CU | CO |
|---|---|---|---|---|---|---|---|---|---|---|---|---|---|---|
| IRISH | CB (2) | 13,784 | **−0.22** | −0.24 | **0.45** | 0.45 | **0.97** | 0.97 | **0.23** | 0.23 | 0.59 | 0.58 | **0.93** | 0.93 |
| | DB (4) | 20,758 | **−0.02** | −0.05 | **0.32** | 0.33 | **0.97** | 0.96 | **−0.27** | −0.33 | **0.57** | 0.61 | **0.92** | 0.92 |
| | ALL (6) | 34,542 | **−0.09** | −0.11 | **0.36** | 0.37 | **0.97** | 0.97 | **−0.10** | −0.15 | **0.58** | 0.60 | **0.92** | 0.92 |
| ECHAN | CB (1) | 17,085 | **0.19** | 0.19 | **0.43** | 0.43 | **0.96** | 0.96 | - | - | - | - | - | - |
| | DB (3) | 21,956 | 0.34 | 0.33 | **0.47** | 0.47 | **0.91** | 0.91 | - | - | - | - | - | - |
| | ALL (4) | 39,041 | 0.30 | 0.29 | **0.46** | 0.46 | 0.93 | 0.92 | - | - | - | - | - | - |
| GOBIS | CB (8) | 95,441 | **−0.01** | −0.01 | **0.37** | 0.37 | **0.96** | 0.96 | **0.34** | 0.45 | **1.03** | 1.11 | **0.89** | 0.89 |
| | DB (12) | 111,202 | **0.03** | 0.04 | 0.36 | 0.35 | **0.97** | 0.96 | −0.45 | −0.33 | **1.12** | 1.20 | **0.88** | 0.87 |
| | ALL (20) | 206,643 | **0.01** | 0.02 | 0.37 | 0.36 | **0.96** | 0.96 | −0.13 | −0.02 | **1.09** | 1.16 | **0.88** | 0.88 |
| NIBSH | CB (5) | 58,313 | −0.18 | −0.15 | 0.34 | 0.33 | 0.96 | 0.96 | **0.38** | 0.61 | **1.05** | 1.20 | **0.89** | 0.89 |
| | DB (5) | 34,954 | −0.13 | −0.09 | 0.32 | 0.31 | 0.97 | 0.97 | **0.13** | 0.38 | **0.70** | 0.83 | **0.93** | 0.93 |
| | ALL (10) | 93,267 | −0.16 | −0.12 | 0.33 | 0.32 | 0.97 | 0.97 | **0.26** | 0.49 | **0.88** | 1.02 | **0.91** | 0.91 |
| WIBSH | CB (1) | 5294 | −0.12 | −0.11 | **0.38** | 0.38 | **0.96** | 0.96 | −0.13 | −0.03 | **0.64** | 0.66 | **0.92** | 0.92 |
| | DB (4) | 32,258 | **−0.04** | −0.07 | **0.31** | 0.31 | **0.98** | 0.98 | **0.08** | 0.16 | **0.62** | 0.66 | **0.95** | 0.95 |
| | ALL (5) | 37,552 | **−0.06** | −0.07 | **0.32** | 0.33 | **0.98** | 0.97 | **0.04** | 0.12 | **0.62** | 0.66 | **0.94** | 0.94 |
| GIBST | CB (3) | 26,173 | 0.10 | **0.09** | 0.31 | **0.29** | 0.85 | **0.86** | 0.48 | **0.28** | 1.15 | **0.98** | 0.62 | **0.68** |
| | DB (1) | 8126 | **−0.03** | −0.04 | **0.23** | 0.23 | **0.95** | 0.95 | **−0.07** | −0.09 | **0.47** | 0.47 | 0.84 | **0.85** |
| | ALL (4) | 34,299 | 0.07 | 0.06 | 0.29 | 0.28 | 0.88 | 0.89 | 0.34 | 0.18 | 0.98 | 0.85 | 0.68 | 0.72 |
| CADIZ | CB (1) | 8743 | 0.32 | **0.27** | 0.51 | **0.47** | 0.77 | **0.79** | 1.05 | **0.65** | 1.76 | **1.45** | 0.58 | **0.61** |
| | DB (2) | 14,159 | **−0.04** | −0.10 | **0.25** | 0.27 | **0.96** | 0.96 | 0.15 | −0.00 | 0.72 | 0.68 | 0.91 | 0.92 |
| | ALL (3) | 22,902 | 0.08 | **0.02** | 0.34 | **0.33** | 0.90 | 0.90 | 0.45 | **0.21** | 1.07 | **0.93** | 0.80 | **0.81** |
| WSMED | CB (4) | 56,832 | **−0.06** | −0.06 | **0.21** | 0.21 | **0.90** | 0.90 | 0.04 | **−0.03** | 0.75 | **0.70** | 0.71 | **0.74** |
| | DB (7) | 33,184 | **−0.13** | −0.14 | **0.29** | 0.29 | **0.93** | 0.93 | **−0.21** | −0.24 | 0.59 | 0.58 | 0.83 | 0.84 |
| | ALL (11) | 90,016 | **−0.11** | −0.11 | **0.26** | 0.26 | **0.92** | 0.92 | **−0.12** | −0.16 | 0.65 | **0.63** | 0.78 | **0.80** |
| ICANA | CB (1) | 6767 | **−0.25** | −0.26 | **0.31** | 0.32 | **0.84** | 0.84 | **−0.17** | −0.27 | **1.09** | 1.12 | **0.35** | 0.33 |
| | DB (2) | 15,857 | **−0.16** | −0.18 | **0.24** | 0.26 | **0.92** | 0.92 | 0.28 | 0.23 | **0.86** | 0.87 | **0.66** | 0.66 |
| | ALL (3) | 22,624 | **−0.19** | −0.21 | **0.26** | 0.28 | **0.89** | 0.89 | 0.13 | 0.07 | **0.94** | 0.95 | **0.56** | 0.55 |
| TOTAL | CB (18) | 186,928 | **−0.05** | −0.05 | **0.34** | 0.34 | **0.93** | 0.93 | **0.24** | 0.27 | **0.93** | 0.95 | 0.81 | 0.81 |
| | DB (31) | 228,392 | **−0.03** | −0.03 | **0.32** | 0.32 | **0.95** | 0.95 | −0.29 | −0.27 | **0.87** | 0.90 | 0.86 | 0.86 |
| | ALL (49) | 422,451 | **−0.03** | −0.04 | **0.33** | 0.33 | **0.94** | 0.94 | −0.10 | −0.07 | **0.89** | 0.92 | 0.84 | 0.84 |

**Table 3.** Error metrics for the test runs with and without current–wave coupling (IBI-CU and IBI-CO test runs, respectively) computed with satellite observations. Time period: 2018. Variable: Significant Wave Height (SWH). Metrics computed for the whole IBI service domain and for the 9 Validation regions (i.e., IRISH, ECHAN, GOBIS, NIBSH, WIBSH, GIBST, CADIZ, WSMED, ICANA) used by the Copernicus Marine IBI-MFC service. Metrics are computed using the available L3 CMEMS altimeter data (Janson-2, Janson-3, Saral, Cryosat-2 and Sentinel3) and HY-2A satellite data. Each error metric (Bias, Root-Mean-Square differences (RMSD) Correlation (CCOR) and Scatter Index (SI2)) provided for each model solution. N counts the size of available sample after the SWH data pre-process. Bold numbers highlight the best performing dataset.

| | | SWH | | | | | | | | |
| | | N | | BIAS (m) | | RMSD (m) | | CCOR | | SI2 (%) | |
| | | IBI-CU | IBI-CO | IBI-CU | IBI-CO | IBI-CU | IBI-CO | IBI-CU | IBI-CO | IBI-CU | IBI_CO |
|---|---|---|---|---|---|---|---|---|---|---|---|
| IRISH | CMEMS L3 | 11,265 | 11,206 | **−0.10** | −0.11 | **0.34** | 0.35 | **0.98** | 0.98 | **12.27** | 11.43 |
| | HY-2A | 2893 | 2890 | **−0.05** | −0.07 | **0.42** | 0.43 | **0.96** | 0.96 | **15.27** | 15.41 |
| ECHAN | CMEMS L3 | 8992 | 8996 | **−0.14** | −0.14 | **0.35** | 0.36 | **0.93** | 0.93 | **19.64** | 19.91 |
| | HY-2A | 2172 | 2167 | **−0.08** | −0.09 | **0.38** | 0.39 | **0.92** | 0.92 | **22.01** | 22.18 |
| GOBIS | CMEMS L3 | 36,110 | 36,074 | **−0.07** | −0.08 | **0.30** | 0.31 | **0.98** | 0.98 | **11.30** | 11.48 |
| | HY-2A | 9578 | 9574 | **−0.00** | −0.01 | **0.32** | 0.33 | **0.98** | 0.98 | **12.37** | 12.50 |
| NIBSH | CMEMS L3 | 5411 | 5411 | −0.10 | **−0.08** | 0.29 | 0.28 | **0.98** | 0.98 | **11.39** | 11.47 |
| | HY-2A | 1482 | 1482 | −0.02 | **0.00** | **0.30** | 0.30 | **0.98** | 0.98 | **12.84** | 12.91 |
| WIBSH | CMEMS L3 | 9154 | 9151 | **−0.07** | −0.09 | **0.30** | 0.32 | **0.98** | 0.97 | **10.81** | 11.15 |
| | HY-2A | 2409 | 2409 | **−0.02** | −0.04 | **0.31** | 0.32 | **0.98** | 0.97 | **11.52** | 11.75 |
| GIBST | CMEMS L3 | 3051 | 3029 | **−0.10** | −0.12 | **0.29** | 0.29 | **0.93** | 0.93 | **18.58** | 18.65 |
| | HY-2A | 569 | 566 | **−0.02** | −0.04 | **0.36** | 0.37 | **0.90** | 0.90 | 25.57 | **25.56** |
| CADIZ | CMEMS L3 | 8288 | 8238 | **−0.12** | −0.13 | **0.27** | 0.29 | **0.97** | 0.97 | **11.95** | 12.32 |
| | HY-2A | 2004 | 1999 | **−0.04** | −0.05 | **0.31** | 0.31 | **0.96** | 0.96 | **14.31** | 14.52 |
| WSMED | CMEMS L3 | 14,073 | 14,047 | **−0.20** | −0.20 | **0.33** | 0.33 | **0.94** | 0.94 | 18.38 | **18.31** |
| | HY-2A | 3027 | 3024 | **−0.13** | −0.13 | **0.35** | 0.35 | 0.93 | 0.93 | 21.67 | **21.64** |
| ICANA | CMEMS L3 | 35,047 | 35,022 | **−0.09** | −0.12 | **0.24** | 0.26 | **0.97** | 0.96 | **10.49** | 10.63 |
| | HY-2A | 8557 | 8554 | **−0.02** | −0.05 | 0.25 | **0.24** | **0.96** | 0.95 | **12.07** | 12.30 |
| TOTAL | CMEMS L3 | 221,521 | 221,321 | **−0.09** | −0.11 | **0.30** | 0.31 | **0.98** | 0.98 | **10.94** | 11.05 |
| | HY-2A | 57,865 | 57,845 | **−0.02** | −0.04 | **0.32** | 0.32 | **0.98** | 0.98 | **12.15** | 12.28 |

Predictably, current coupling has no influence in severe storms, where strong wind forcings control the wave model output, with good accuracy in both cases: IBI-CU and IBI-CO solutions. Figure 4 shows the time series at locations for the three biggest storms in western IBI area in 2018: Carmen, the 1 January in the Cantabrian Sea, Emma, the 28 February in the Gulf of Cadiz and Ali, the 19 September on the Irish coast. In these storms, current refraction has a limited impact on wave height patterns, with results (IBI-CU and IBI-CO) more similar than the usual state.

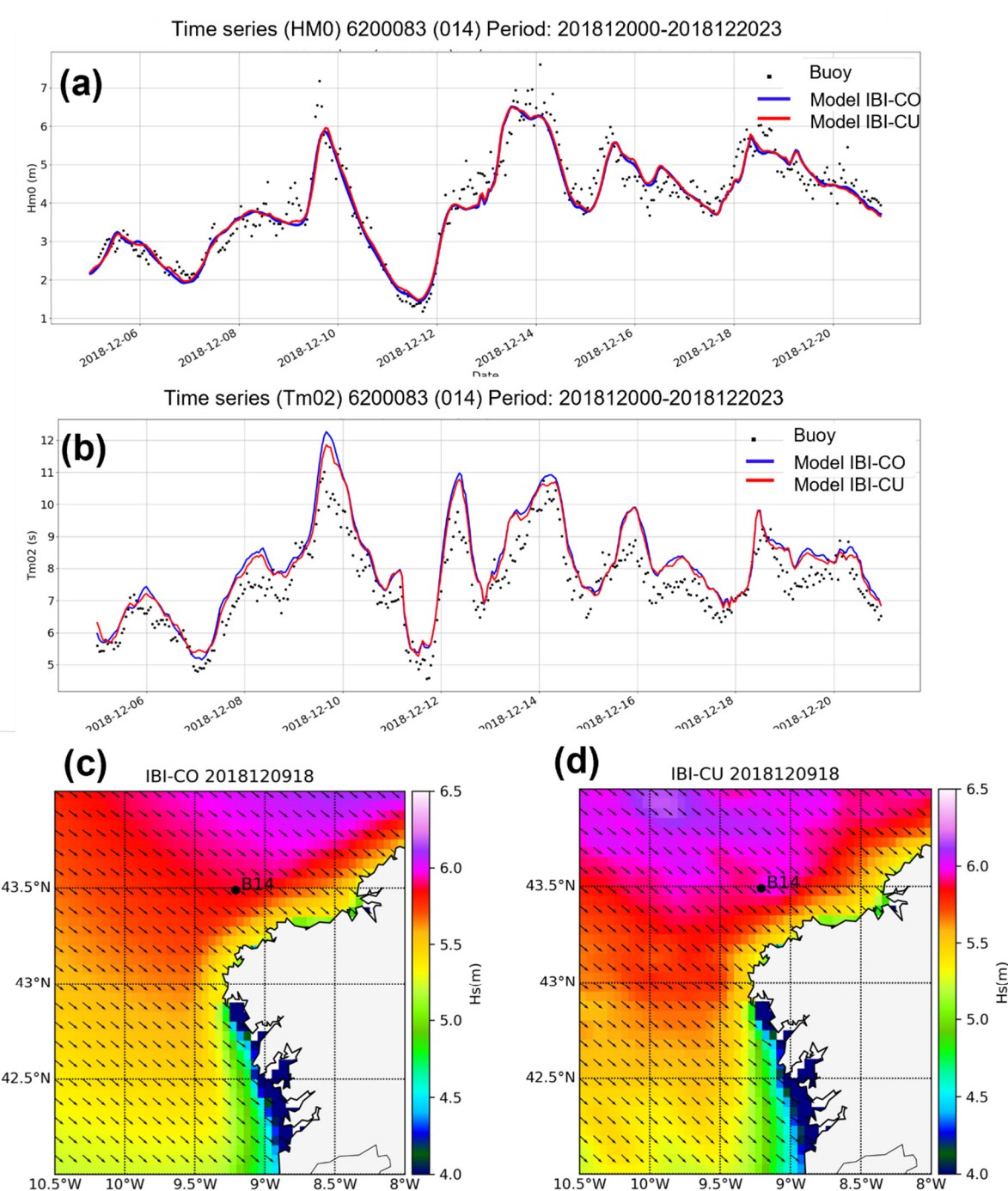

**Figure 3.** Typical time series of significant wave height (**a**) and mean period (**b**) at the buoy 6200083 for a period of 2 weeks (5–12 December 2018). The observed values are represented by the black dots. Two model results are shown, one including current coupling (red line, IBI-CU) and the other without currents (blue, IBI-CO). On the bottom, example of the modeled situation at 18:00 UTC 9 December 2018 for the western coast of Galicia for the coupled model, IBI-CU (**d**) and no coupled model, IBI-CO (**c**). Point B14 is the location of the buoy 6200083.

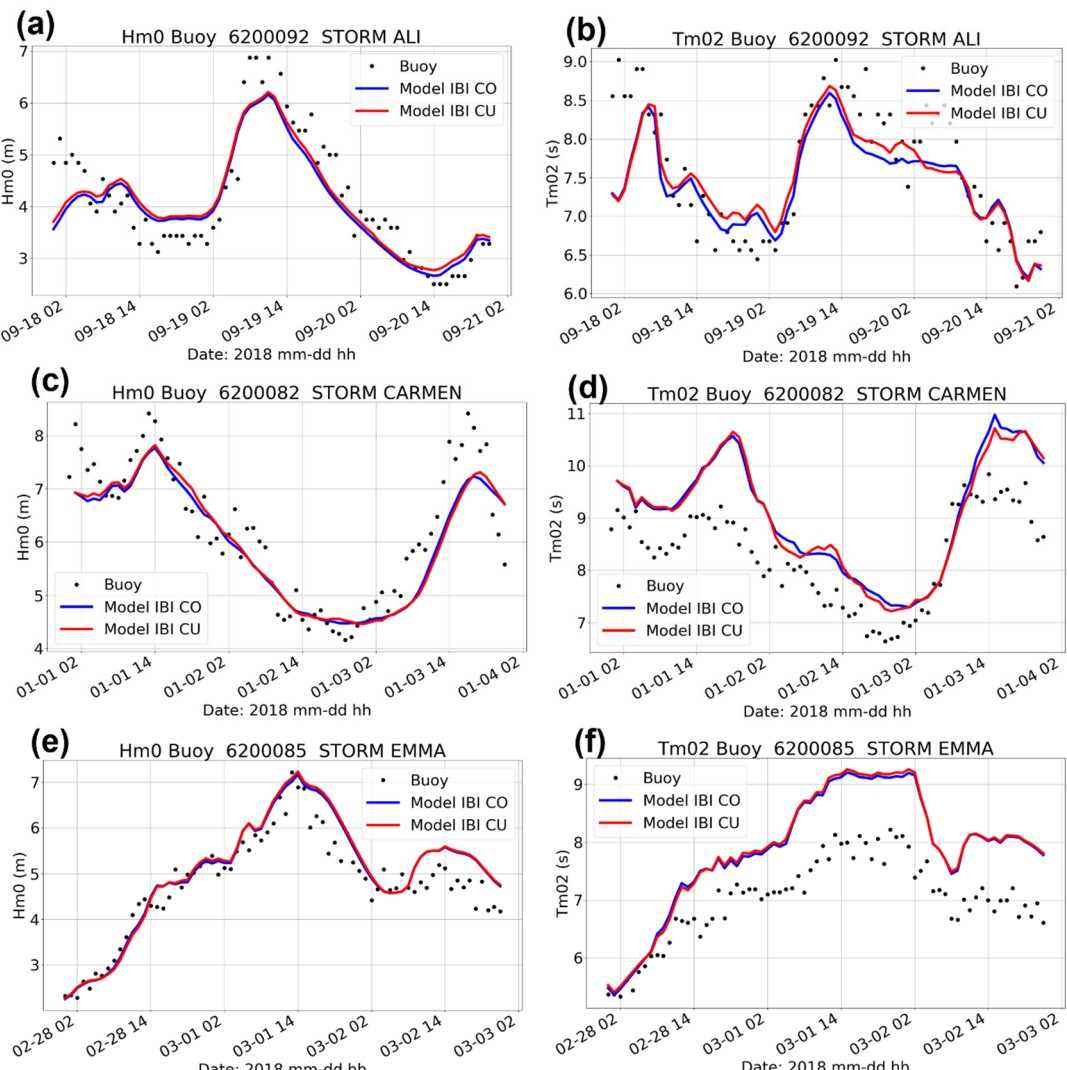

**Figure 4.** Comparison of coupled model IBI-CU, (red line) and no coupled, IBI-CO (blue line) against mooring buoys for different Storms: Ali ((**a**) Hs and (**b**) $Tm_{02}$ at buoy 620092 on Irish coast), Carmen ((**c**) *Hs*, (**d**)$Tm_{02}$ at the buoy 6200082, Gijon coastal buoy at Cantabrian Sea) and Emma ((**e**) *Hs*, (**f**) $Tm_{02}$ at buoy 620085 in Gulf of Cadiz). The observed values are represented by the black dots.

*3.2. Evaluation of Data Assimilation Performance: Validation of New IBI Wave Analysis*

Data available for the year 2018 were assimilated to produce the hindcast. Analyses were performed every hour with a data window of 3 h (i.e., data 1.5 h either side of the analysis time were assimilated). A consequence of this arrangement is that the data are not all independent between analyses; however, in testing, we found that this performed better than insisting on the strict independence of the data. This is because with such a short analysis time step, a narrower window does not always contain enough data for an effective analysis. It also leads to a kind of smoothing of the data in time, which could potentially benefit the analysis because, if the data are too different between analyses, and lead to too big a disparity between observation and model in an ensuing analysis step, the data could be rejected. These are not general principles in data assimilation and depend on the assimilation algorithm used. Because optimal interpolation treats all data assimilated for a given analysis as being observed at the same time as the analysis, the correlation of data between analyses is not a concern, as it would be for a time-dependent assimilation algorithm, such as the Kalman filter.

For comparison, the 2018 model runs were also performed without data assimilation. We refer to the runs with data assimilation and no currents as the IBI-DA runs, and those without as the reference runs (IBI-CO).

Bulk statistics for the entire 2018 study period show that the model is closer to the observations with data assimilated than without. These results are summarized in Table 4. Interestingly, the bias, while shrinking in magnitude, changes from being negative to being positive with the assimilation of data. On the other hand, the scatter indices are reduced by DA in the whole IBI region, particularly in CADIZ subregion, where SI drops by 1.8%. In this area, however, the impact of DA on the model during storm Emma (26 February–7 March 2018) is more like the whole IBI region.

**Table 4.** Biases and scatter indices for control run (IBI-CO) and Data Assimilation simulation (IBI-DA) for the whole IBI area and the CADIZ subregion. In the last case, metrics computed not only for the year 2018, but specifically for the period of the Emma storm (26 February–7 March 2018).

| | SWH | | | | | |
|---|---|---|---|---|---|---|
| | N | | BIAS (m) | | SI | |
| | IBI-CO | IBI-DA | IBI-CO | IBI-DA | IBI-CO | IBI-DA |
| TOTAL IBI (2018) | 58,972 | 59,027 | −0.05 | 0.02 | 12.2 | 11.2 |
| CADIZ (2018) | 7651 | 7656 | −0.04 | 0.03 | 11.9 | 10.1 |
| CADIZ (Storm Emma) | 212 | 212 | 0.21 | 0.23 | 8.92 | 7.99 |

The scatter plots in Figure 5 gather all the observation–model data pairs. They were generated for the entire two-year period and allow us to examine the validation in more fine-grained detail. The IBI-DA run is visually more concentrated about the center of mass, the black line representing a one-to-one correspondence between the observations and model. The colored squares on that axis also appear hotter in color, indicating that more data pairs are concentrated on it. Finally, the linear regression of the IBI-DA run (the red line) has a gradient closer to one than that of the reference run, though again, for very low Hs, the regression line of the reference run is closer to the center of mass.

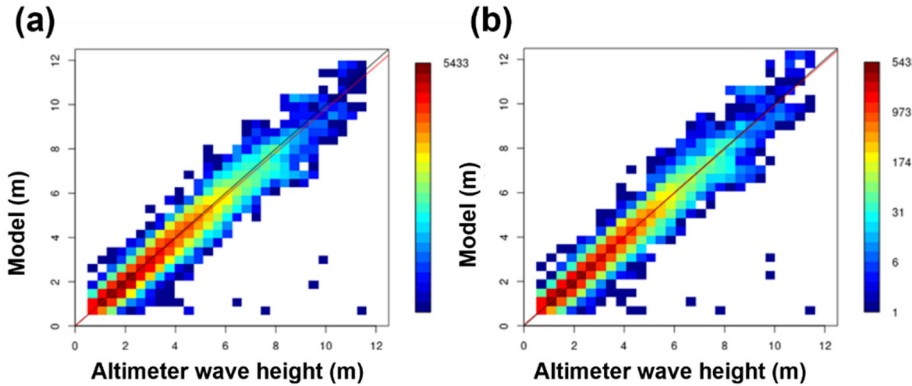

**Figure 5.** Density scatter plots comparing observed Hs against modelled Hs for all of 2018. The control run (IBI-CO, (**b**)) is on the left, the IBI-DA ((**a**)) run on the right. Each box represents the points found within the range of Hs it covers, and the color scale indicates the number of validated points within the box. The red line is a linear regression, with the black line representing an ideal 1-1 correspondence between model and data.

The quantile–quantile plots in Figure 6 allow us to compare the statistical distributions of the observed and modelled *Hs*, without and with data assimilation. These plots were produced using only the model points that correspond to observations. Assuming reliable data, a straight line of gradient equal to one would imply that the model produces an

identical distribution of *Hs* to the observations. Comparing the plot for the IBI-DA run with that of the control run, we see that data assimilation helps bring the model's Hs distribution closer to that of the observations for most of the range of *Hs*. This is especially true for *Hs* between 6 and 9 m. For low *Hs*, up to around 2 m, the IBI-DA run's representativity is slightly worse, but the difference is small, and these wave heights are of much less interest to seafarers, so the inaccuracy here can be forgiven. At extremely high Hs, the distribution of the IBI-DA run is skewed high. There are not many data in this extreme regime to begin with, and the control run already suggests this by oscillating around the ideal unit gradient. Furthermore, the modelling of extremely high waves is even less reliable, though we should be cautious, given the paucity of data in the regime. One possible explanation for this over-correction could be because the reference model is under-estimating moderately high *Hs*, which are great in number, so when the analysis corrects these upward, it inadvertently increases the extremely high Hs as well. In other words, the moderately high *Hs*, because of their larger number, are weighting the analysis more than the extreme *Hs*. In the simple OI data assimilation scheme implemented here, where model errors are constant and covariances are defined solely based on the distance between points, there is no way for it to selectively apply the correction, in such a way as to avoid incorrectly increasing these extreme wave heights.

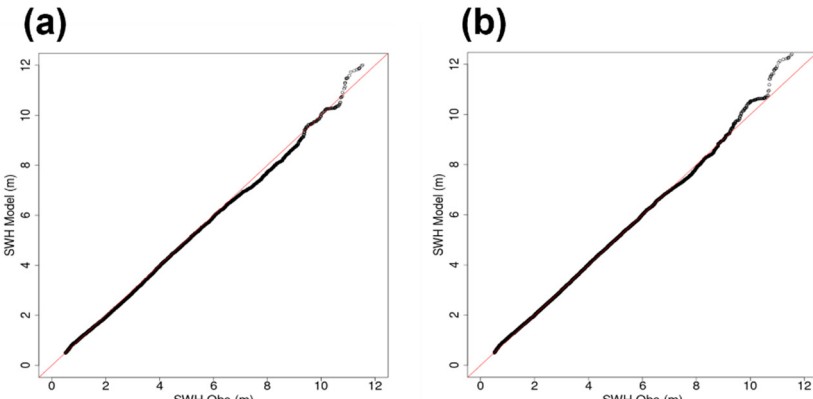

**Figure 6.** Quantile–quantile plots of observed Significant Wave Height (SWH), *Hs*, against modelled *Hs* for all of 2018. On the left is the IBI-CO run (**a**), on the right the IBI-DA run (**b**). The red line is the unit gradient line.

The bar plots for the monthly means in Figure 7 show, briefly, a consistent reduction in scatter index with data assimilated; it is most reduced in December, March and April, and least in January and September. The bar plots for bias reflect the shift from negative overall bias to positive overall bias, but in some instances, the absolute bias is greater with data assimilation—especially so in July, where the control run's bias is already positive. The fact that the data assimilation always results in the bias tending positive suggests that a bias remains in the data. The monthly diagnostics reveals a moderate seasonal signature in scatter index, with higher scatter indices in the summer months, conserved in the IBI-DA run. The seasonal signature for the bias is almost inverted in the DA run, with higher absolute biases in summer than in winter (apart from November and February). With all this said, the highest absolute bias for all months in the IBI-DA run is only just over 5 cm, which, to put it into perspective, is about the same as the bias for the control run for the whole year.

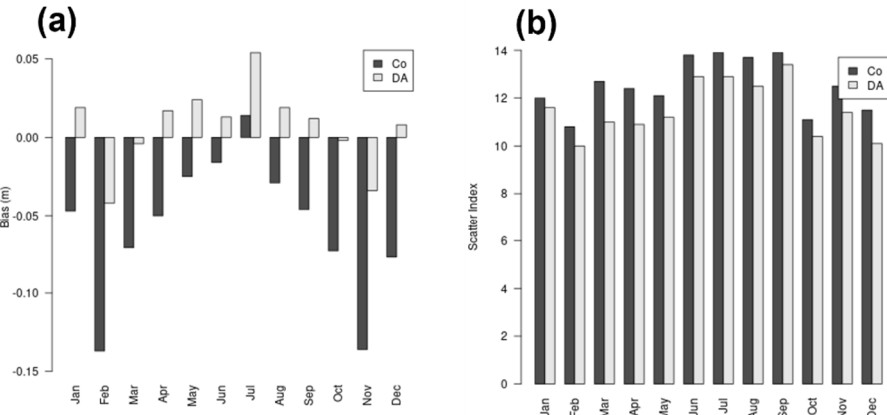

**Figure 7.** Model (IBI-DA and IBI-CO) Bias (**a**) and Scatter Index (**b**) compared with HY-2A altimeter Hs observations.

## 4. Discussion

In the previous section, the main results from the IBI wave model sensitivity tests performed (i.e., IBI-CU and IBI-DA) are shown. All the IBI wave simulations are validated with observational data sources for significant wave height and period, and the two test runs are compared with the control run (IBI-CO), which is performed using the same model set-up that has been used in the IBI operations since 2018.

The main aim of these sensitivity tests is to assess the impacts that potential upgrades in the wave model set-up have on the IBI wave model solution. The scientific qualification performed (based on comparisons of the IBI test model solutions with the control run, and the available observations) should provide enough information to decide if some of the proposed set-up novelties may be included in a new operational IBI-MFC wave system release. To make a decision on the adequateness of the proposed set-up upgrades to be launched as part of the next IBI operational wave system release, it would be necessary to verify if the new model solution improves or degrades, with respect to the control (and at what level).

Overall, skill computed along the whole test year do not show substantial improvement with wave–current coupling implementation, in terms of significant wave height accuracy. Model coupling performance (IBI-CU) is a little better for the mean wave period, but in any case, for both parameters, Hs and $Tm_{02}$, the improvement with wave–current coupling is small and does not exceed 0.2%. The scatter index *SI*2 for the significant wave height (Table 3) and the 3% RMSD for the mean wave period computed with hourly observations at mooring buoys (Table 2). However, as it is well known, and extensively described in the literature [43–48], there are very specific situations when this current forcing improves the wave simulation (mostly in shallower coastal waters and related to the same case, where local waves propagate in the opposite direction to prevailing surface currents) differences on mean period have been observed, reducing the quality of the ocean-coupling model. There are some special cases, marked by differences, mainly in the mean period, representing a decrease in the wave model skill when coupling is activated, that should be discussed.

For instance, even though current forcing increases values of wave height so metrics are slightly better for the coupled model (IBI-CU), we see how, in some regions, such as the Gulf of Cadiz and Strait of Gibraltar (CADIZ and GIBST region), a significant decrease in quality in the coupled simulation (and particularly in the period) is detected.

This is seen in the two coastal buoys located close to the strait of Gibraltar (buoys 6101404 and Tarifa-coast-buoy), where the difficulty of obtaining a good ocean current performance produces worse results: they are strongly influenced by an unrealistic ocean current overestimate, the $Tm_{02}$, when *Hs* is low (less than 0.5 m). Figure 8 illustrates this pattern at buoy 6200085 (Cadiz deep buoy; see locations in the maps depicted in Figure 2)

for a whole month, September 2018. This case is used because of the availability of surface current observations (the buoy location is covered by the Puertos del Estado HF Radar System) [49]. Skill metrics for mean period $Tm_{02}$ weaken in the coupled model (correlation decreased from 0.83 for control simulation to 0.74 for IBI-CU in September). The time series depicted in Figure 8, help in analyzing this decrease in the validation scores. Figure 8b shows how differences in the wave period between the coupled and control run mainly occur on some specific days (events depicted in the figure by the dark blue squares). It is important to note that these days, when overestimated values for $Tm_{02}$ in the IBI-CU solution are identified, show very low significant wave height values (Hs values being lower than 0.5 m; see Figure 8a). The IBI model surface currents can be validated with HF Radar observation for the day 3 September 2018. Comparison of daily averaged IBI model currents (Figure 8c) with the HF radar observed ones (Figure 8d) show how the unrealistic simulation of surface current at this location (P16 point indicates location of Buoy 6200085 used for the wave validation) spuriously introduces energy into the wave model, generating the increments of mean period in these cases, marked by low wave heights.

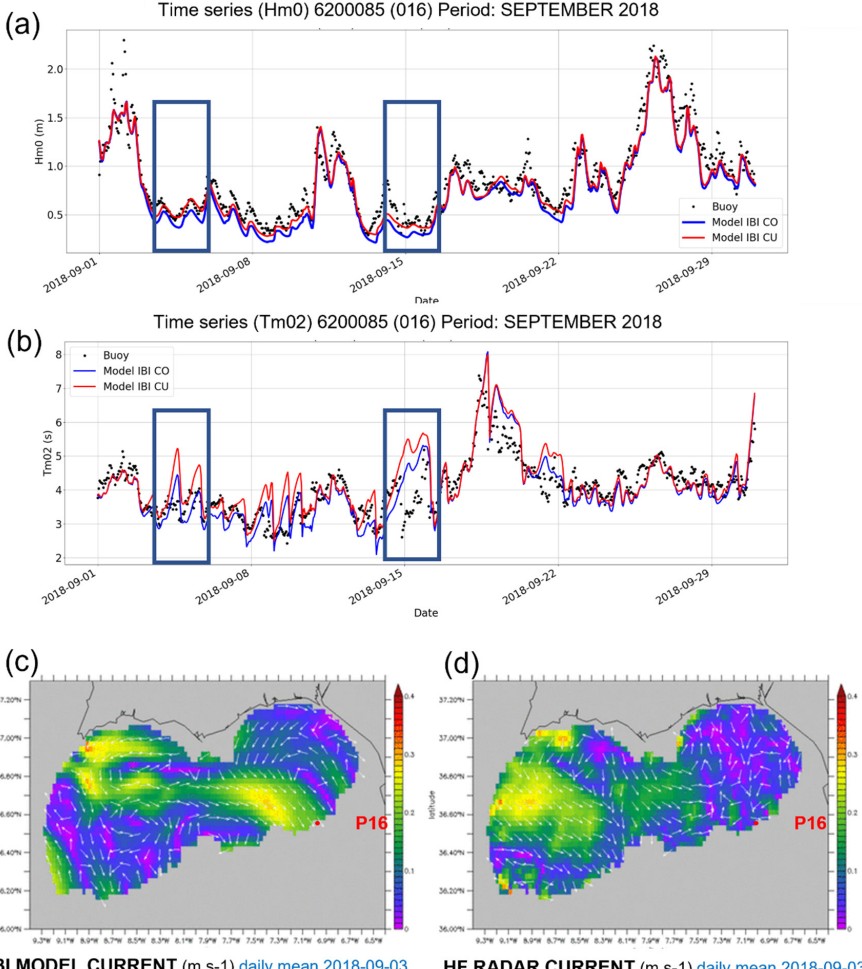

**Figure 8.** Monthly time series for *Hs* (**a**) and $Tm_{02}$ (**b**) for the simulations of coupled model IBI-CU, (red line) and no coupled, IBI-CO (blue line) at buoy 6200085 in Gulf of Cadiz, observed data in black dots. The square in dark blue square encloses the overestimated values for $Tm_{02}$ in IBI-CU solution. On the bottom, images of the validation tool NARVAL (North Atlantic Regional Validation; [50,51]): daily mean of current velocity and direction for the IBI model solution (**c**) and HF radar (**d**). Point P16 is the location of the buoy 6200085.

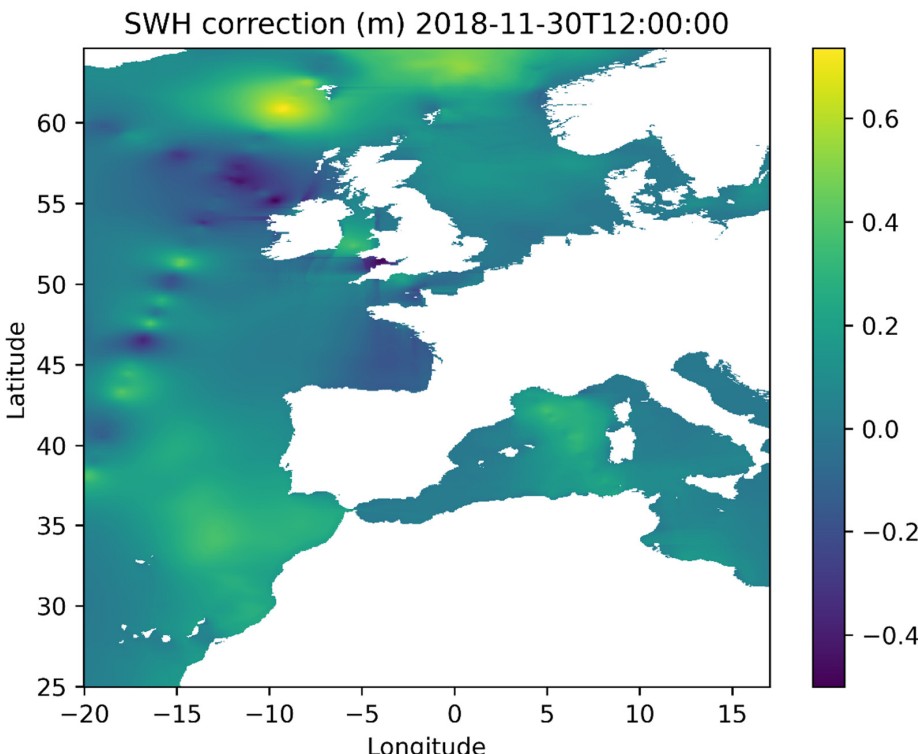

**Figure 9.** Difference of Significant Wave Height, Hs, (in meters) from IBI-DA and IBI-CO runs during a North-East Atlantic storm (on the 30 November 2018 at 12 UTC).

These results emphasize the importance, when coupling ocean current with waves, of having realistic high-quality model surface current fields. As shown in this example, the use of model currents inputs, locally affected by unrealistic model features, may spoil the wave model performance, especially in very low energetic situations outside of the main storm events. As such, ref. [52] shows a similar misfit induced by model currents in Southern Ocean.

The assimilation of the altimeter data showed a significant reduction in the bias and scatter index on significant wave height. On average, the scatter index of Hs is improved by roughly 8% in open ocean. We noticed that the assimilation is skilled to efficiently correct the wave model errors related to the uncertainties of the wind forcing in the North Atlantic, especially during storm events, as illustrated in Figure 9. This clearly brings better initial conditions for swell propagation to coastal areas, as revealed at Cadiz buoy. However, there is still room for improvement in the assimilation scheme, namely better estimates of the covariance model errors, by taking into account the variability of the sea state in the IBI domain. Moreover, the use of variable correlation length, depending on whether the sea state is wind sea or swell dominated, will induce a better spread of the assimilation correction on the model grid points.

Results from the new IBI wave systems (proposed to be the new IBI wave operational release, named here IBI-OP) can be seen in Table 5.

**Table 5.** Estimated Accuracy Numbers related to Significant Wave Height (year 2018) for the IBI wave model scenarios. On the right (column IBI-OP), the new IBI operational wave system. Observational source reference: HY-2A satellite-derived product.

| | TOTAL IBI SWH | | | |
| --- | --- | --- | --- | --- |
| | **IBI-CO** | **IBI-CU** | **IBI-DA** | **IBI-OP** |
| N | 57,845 | 57,865 | 57,899 | 57,921 |
| BIAS | −0.04 | −0.02 | 0.02 | 0.03 |
| CORR | 0.98 | 0.98 | 0.98 | 0.98 |
| RMSE | 0.32 | 0.32 | 0.29 | 0.29 |
| HH (%) | 10.95 | 10.72 | 9.73 | 9.67 |
| SI2 (%) | 12.28 | 12.15 | 11.13 | 11.14 |

The statistical comparison for the entire year 2018 of the four scenarios, defined in Table 1, and the HY-2A altimeter shows a good evolution for each model upgrade (Table 5). However, the most relevant improvement is due to the data assimilation implementation (IBI-DA), with a scatter index *SI*2 and error indicator *HH,* decreasing more than 1%. The impact of ocean current, although less significant, improves the model qualification, increasing the significant wave height and with better accuracy of mean period (Tables 2 and 3).

## 5. Conclusions

Two major conclusions emerge from these tests. First, the significant improvement on the wave model set-up that was used in the IBI operations in December 2019 (IBI-CO), activating current forcing (IBI-CU), data assimilation (IBI-DA), or both (IBI-OP), with the major upgrade in the performances, including data assimilation, with the scatter error indicator decreasing by more than 1% (Table 5). Second, the importance of having realistic high-quality forcings fields. In particular, the use of unrealistic currents inputs may spoil the wave model performance (Figure 8).

The research performed, combining information from different wave model solutions and several observational data sources, benefited the IBI-MFC operational wave forecast service, supporting the decision-making process, related to their latest major operational upgrade (occurred in the Copernicus Marine Operational Release March 2020). The extensive assessment performed with the IBI-DA and IBI-CU run, presented here, was useful to demonstrate, for each proposed IBI model set-up novelty, the significance of the associated improvements (in terms of IBI wave forecasting local added value). After verifying the impacts that these two proposed novelties have on the IBI wave solution (and always using, as a referential solution, the IBI wave model configuration in operations), a new IBI wave configuration (combining both the data assimilation and the current–wave coupling) were prepared to be transitioned into operations.

The significant improvement achieved by this IBI-OP configuration, with respect to the control one (IBI-CO, the IBI set-up currently in operations), in terms of wave solution over the IBI area, throughout the year 2018, supports the IBI-MFC decision to use this set-up for operational purposes, to be implemented in the IBI Wave, operational from March 2020. Furthermore, similar performance metrics were achieved by the systems in operations (see information from a 2-year validation in the Quality Information Document of the Copernicus Marine IBI wave product, Toledano et al. 2021 [31]). Finally, note that this research was useful not only to evolve the IBI-MFC operational wave forecast service, but also has been helpful to open new working lines prioritized in the Copernicus Marine service roadmap:

- Currently, the IBI-MFC is in the process of enhancing the coupling, in both directions, between waves and ocean dynamics. In the December 2020 release, the upgraded IBI wave system included improved computations of coupling parameters, such as the surface stress, and wave-breaking-induced turbulence in the ocean mixed layer. These parameters are used to drive a coupling run with the IBI MFC ocean forecast

run, based on a NEMO model application. The sensitivity coupling tests of using the Stokes–Coriolis forcing, the sea-state-dependent stress momentum fluxes and including wave-breaking energy flux in the vertical mixing, are all validated and the impact on improving the forecast is addressed by [8,53,54].

- With respect to data assimilation, directional wave spectra data assimilation based on the optimal interpolation method, applied to mean wave parameters (total energy and wave number components) of each wave train composing the wave spectrum [55], will be implemented, in order to increase the impact of assimilation [7]. Likewise, assimilation of wave data from swath altimetry is expected to be tested in the future.

**Author Contributions:** Conceptualization, M.G.S., L.A., C.T. and M.G.; methodology, C.T. and M.G.; software, C.T., M.G., A.D. and P.L.; validation, C.T., M.G., P.L. and A.D.; formal analysis, C.T., M.G., L.A. and M.G.S.; investigation, C.T. and M.G.; data curation, C.T., M.G., P.L. and A.D.; writing—original draft preparation, C.T., M.G., M.G.S. and L.A.; supervision, L.A. and M.G.S.; project administration, M.G.S. and L.A. All authors have read and agreed to the published version of the manuscript.

**Funding:** This research was funded in the framework of the E.U. Copernicus Marine Service for the IBI region (CMEMS Contract No. 100-IBI-OPER-CMEMS and Copernicus-2 Marine service contract 21002L6-COP-MFC IBI-5600).

**Institutional Review Board Statement:** Not applicable.

**Informed Consent Statement:** Not applicable.

**Data Availability Statement:** The Copernicus Regional IBI-MFC Wave solution can be downloaded at Copernicus Marine Environmental Monitoring Service (CMEMS) (http://marine.copernicus.eu/, accessed on 18 January 2022).

**Acknowledgments:** This study has been conducted using E.U. Copernicus Marine Service Information. Specifically, from their in-situ TAC Product (INISITU_IBI_NRT_OBSERVATIONS_013_033) and CMEMS L3 wave altimetric product (WAVE_GLO_WAV_l3_SWH_NTRT_OBSERVATIONS_014_001). Likewise, satellite observations from the HY2A mission, provided by the CNES, have been used.

**Conflicts of Interest:** The authors declare no conflict of interest.

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
