# Peer review of "Impacts of an Altimetric Wave Data Assimilation Scheme and Currents-Wave Coupling in an Operational Wave System: The New Copernicus Marine IBI Wave Forecast Service"

_jmse, doi:10.3390/jmse10040457_

Round 1

Reviewer 1 Report

The manuscript considers two improvements to the IBI-MFC wave forecasting system: 1) a sequential OI based DA system and 2) quasi-coupled wave-current interaction. Compared to the baseline model- improvements are found from the DA system, whereas improvements due to the currents is smaller at the observed sites. 

I think that combination of models with observations (i.e. DA) is still the most promesing (and underexplored) operational direction to improve wave forecasts, and manuscripts explorign the potential of wave DA are definitely of interest. Further, a benchmark of the new improved to the old system is usefull for users of the product. I would therefore recommend publishing the article after a revision addressing comments below.

Pieter Smit, Head of Research, Sofar Ocean

Major Comments
- There are quite a few broken references, formatting issues and the like in the manuscript. While this did not prevent me from reviewing it was somewhat distracting/annoying haveing to parse the references from context (e.g. it took me a while to realize that figure 5 is repeated due to a reference error).

- the Kalman Gain expressed in equation 3 confuses me. It basically states that observed points model and observation error are assumed equal - so that only half of the innovation is ever applied if a model and obseration point coincide. It may be that altimeter and model errors are somewhat comparible - but it would be good if this assumption is introduced explicitly.

- I am confused why the authors use a different comparison approach for the DA and CU runs. For the CU runs buoy data is used exlusively- whereas for the DA focus is on the Altimeter data. I would have expected both sets of observations to be used in both cases?

- I would argue that the model setup does not couple wave-current interaction as it is referred to in the traditional sense. The current field is essentially an external forcing field (like the wind) - as there is no feedback from the wave model to the current model. Hence referrign to the system as "coupled" gives the wrong impression.

- The currents appear to have low influence at buoys sites- but there does appear to be a perceptual change at off-shore spatial distribution of waveheights (e.g. Fig 3). Did the authors do any anecdotal comparisons with altimeter data to see if this is better represented in the CU model?

- The coupling is described in section 2.2.2: it is probably usefull to note that action (and not energy) is the conserved quantity for wave current interaction (otherwise technically the work term that radiation stress does against the flow needs to be introduced). Secondly- it is unclear to me if the authors included equation 13, or if it was already in the model. If the former, we would need more verification, if the latter, why call it out so specifically, reference would suffice (no need to intruce eq 13?). It is probably also good to inform the reader why neglecting dd/dt is a good approximation for most of the domain (only relevant for significant tidal motions in shallow water).

Author Response

First, the authors would like to thank both Reviewers for their valuable comments. Following their suggestions, the revised manuscript has been certainly improved.   

The authors agree with reviewer’s #1 suggestions and all the points has been addressed. A pint-by-point response is provided below:  

  • There are quite a few broken references, formatting issues and the like in the manuscript: As suggested, the broken references and formatting issues have been solved.

  • The Kalman Gain expressed in equation 3. It basically states that observed points model and observation error are assumed equal - so that only half of the innovation is ever applied if a model and observation point coincide. It may be that altimeter and model errors are somewhat comparable - but it would be good if this assumption is introduced explicitly:

This was indeed inadvertently omitted. The authors have made this explicit and motivated it by explaining that it is an empirical choice, and further changes have been made to this section so that it flows more logically.

  • Why the authors use a different comparison approach for the DA and CU runs? For the CU runs buoy data is used exclusively- whereas for the DA focus is on the Altimeter data. I would have expected both sets of observations to be used in both cases?:

The authors state that in the scientific qualification performed it was used Altimeter data for both cases: IBI-CU (see table 3) and IBI-DA. However, the relatively minor improvement identified (when looking at the whole domain as a whole for the IBI coupling (IBI-CU), when compare with the details seen in a more exhaustive comparison with local in situ data and the limitation in the number of figures to be included (the paper in its original version already counts with 9) motivated that not all the figures were displayed, being in this case the info with the current coupling at specific buoys displayed, instead of the one from the satellite comparisons. 

  • The model setup does not couple wave-current interaction as it is referred to in the traditional sense. The current field is essentially an external forcing field (like the wind) - as there is no feedback from the wave model to the current model. Hence referring to the system as "coupled" gives the wrong impression:

The authors agree with the Reviewer’s point of view, and the expression “coupling” has been changed to “current forcing”. 

  • The coupling is described in section 2.2.2:

The current forcing implementation, based on equation 13 (Tolman) with the dd/dt neglected, was already included in the MFWAM. The reason for including this formula is due to the absence of information on how is implemented currents forcings on the MFWAM model.

Reviewer 2 Report

In this work the impacts of an altimetric wave data assimilation scheme and currents-wave coupling in an operational wave system was proposed.

The analysis carried out by the authors is extremely interesting. Although the work is interesting, with good English, some adjustments are necessary before the work can be published. All formulas do not seem to meet the requirements of the journal. Almost all the figures look very bad, and often some information is missing, a great deal of reworking has to be done. A workflow may be added at the end of the methods section and before the results to make the work clearer. All bibliographical references in the text do not meet the requirements of the journal. Finally, about 30% of the reference appear to be from the authors underlining a presence of self-citation. Below are some general and specific comments.

General comments

Introduction: some improvement about the Copernicus program can be added 

Material and methods. Please first of all define and justify the selection of the study area so that the reader can understand where and why the survey is taking place. After that describe the materials that were used and after that how the authors combine the materials in the procedure in order to obtain the results. To make more clear authors must make different paragraph like: 2.1 Study area; 2.2 Copernicus Marine data; 2.3 Ground reference data - mooring buoys…

A workflow may be added at the end of the methods section and before the results to make the work clearer.

Figure quality is very low, all the chapter are different. Is not acceptable for a journal such as Marine Science and Engineering.

Conclusions: Please add the conclusion paragraph.

Specific comments

Line 17: Define MFWAM

Line 41: Define the wave model as WAM, SWaM….

Line 185: check “¡Error! No se encuentra el origen de la referencia”

Line 322: Figure 2, what is the reference system of the figure? Also the north arrow is still missing.

Line 427: Figure 3. What is the reference system in figure 3c and 3d?

Line 435: Figure 4: the text is too small and the quality very poor. Please improve.

Line 470-480: Why does the figure appear twice? please fix

Line 480: Figure5, Please add a letter for each figure i.e. a) and b).

Line 507: Figure 6, Please add a letter for each figure i.e. a) and b).

Line 524: Figure 7, Please add a letter for each figure i.e. a) and b). quality need to be improved. Moreover add in the caption the code meaning (Co and DA).

Line 581: Figure 8. the text is too small, practically unreadable. What is the reference system in figure 8c and 8d?

Line 564: check “¡Error! No se encuentra el origen de la referencia”

Line 570: check “¡Error! No se encuentra el origen de la referencia”

Line 571: check “¡Error! No se encuentra el origen de la referencia”

Line 605: Figure 9 is really bad in terms of quality, add the title of the axes. please improve.

Line 622: check “¡Error! No se encuentra el origen de la referencia”

The following reference can be added to the paper:

  • Copernicus marine service “Le Traon, P. Y., Reppucci, A., Alvarez Fanjul, E., Aouf, L., Behrens, A., Belmonte, M., ... & Zacharioudaki, A. (2019). From observation to information and users: The Copernicus Marine Service perspective. Frontiers in Marine Science, 6, 234.
  • Copernicus marine service. “von Schuckmann, K., Le Traon, P. Y., Smith, N., Pascual, A., Brasseur, P., Fennel, K., ... & Zuo, H. (2018). Copernicus marine service ocean state report. Journal of Operational Oceanography, 11(sup1), S1-S142.”
  • Copernicus data to support European strategy. “Sarvia, F., De Petris, S., & Borgogno-Mondino, E. (2022). Mapping Ecological Focus Areas within the EU CAP Controls Framework by Copernicus Sentinel-2 Data. Agronomy, 12(2), 406.”
  • Copernicus data to support European strategy “Aschbacher, J. (2017). ESA’s earth observation strategy and Copernicus. In Satellite earth observations and their impact on society and policy (pp. 81-86). Springer, Singapore..”

Author Response

First, the authors would like to thank both Reviewers for their valuable comments. Following their suggestions, the revised manuscript has been certainly improved.

The authors agree with reviewer’s #2 suggestions and all the points has been addressed. Formulas, figures, references, and the structure of the text have been updated. As requested, new bibliographical references have been added to the revised manuscript, and equations with punctuation are included now as a regular text. All figures are now uploaded (in tiff files) with the quality requested by the journal.

 A point-by-point response is provided below:

(General comments)

  • Introduction: some improvements about Copernicus program can be added:

As suggested, a new explanation of the global context of Copernicus Programme, in which Marine Copernicus Service is embedded has been added

  • Material and methods. Please first of all define and justify the selection of the study area so that the reader can understand where and why the survey is taking place. After that describe the materials that were used and after that how the authors combine the materials in the procedure in order to obtain the results. To make more clear authors must make different paragraph like: 2.1 Study area; 2.2 Copernicus Marine data; 2.3 Ground reference data - mooring buoys…

The authors agree with this Reviewer’s suggestion, and the structure of section 2 has been changed in the modified manuscript following the recommendation. For sake of clarity some subsections names have been modified, resulting in: 

2.1. The IBI area and IBI-MFC wave model  

2.2. Model sensitivity tests: the proposed IBI Wave model upgraded 

2.2.1. The Altimetric wave DA scheme proposed for IBI 

2.2.2. Wave-current coupling proposed for IBI 

2.2.3. Assessment of Model runs: Evaluation criteria against insitu and altimeter data. 

As suggested, definition of study area included firstly (lines128-130); Methods should be described in section 2 (so, section 2.2 is maintained there) and the title of the last sub-section proposed has also been changed to show that buoy and satellite observational data sources are here described. 

  • A workflow may be added at the end of the methods section and before the results to make the work clearer.

The authors agree with the reviewer and the suggested workflow is now provided in the revised manuscript (prologue of methods section ; table 1).  

  • Conclusions: Please add the conclusion paragraph:

The authors agree with the reviewer and a conclusion section has been added in the revised manuscript. 

(Specific comments)

  • Define MFWAM (line 19):

As suggested, the acronym of MFWAM has been added.

  • Define WAM, SWAN…

As requested, acronyms and bibliographical references describing these wave models have been added.

  • Reference problem in line118:

Solved.

  • Figures 2, 3:

As suggested, geographical reference system (longitude coordinates with West symbol and latitude coordinates with North symbol) are added.

  • Figure 4:

As suggested, the quality have been improved.

  • Figures 5, 6:

Letters for the Figures (a) and b)) have been added.

  • Figure 7, Please add a letter for each figure i.e. a) and b). quality need to be improved. Moreover, add in the caption the code meaning (Co and DA):

Suggested changes are included in revised manuscript.

  • Reference problems (lines 564,570,571 and 622):

The cross-references have been added.

  • The following reference can be added to the paper:

Copernicus marine service “Le Traon, P. Y., Reppucci, A., Alvarez Fanjul, E., Aouf, L., Behrens, A., Belmonte, M., ... & Zacharioudaki, A. (2019). From observation to information and users: The Copernicus Marine Service perspective. Frontiers in Marine Science, 6, 234.

Copernicus marine service. “Von Schuckmann, K., Le Traon, P. Y., Smith, N., Pascual, A., Brasseur, P., Fennel, K., ... & Zuo, H. (2018). Copernicus marine service ocean state report. Journal of Operational Oceanography, 11(sup1), S1-S142.”

Copernicus data to support European strategy. “Sarvia, F., De Petris, S., & Borgogno-Mondino, E. (2022). Mapping Ecological Focus Areas within the EU CAP Controls Framework by Copernicus Sentinel-2 Data. Agronomy, 12(2), 406.”

Copernicus data to support European strategy “Aschbacher, J. (2017). ESA’s earth observation strategy and Copernicus. In Satellite earth observations and their impact on society and policy (pp. 81-86). Springer, Singapore..”

The authors agree with the reviewer’s suggestion and all the pointed references (together with some extra ones) have been added, improving the provided bibliography. 

Round 2

Reviewer 2 Report

The work seems to be definitely improved, nevertheless, some small changes should be made before proceeding with the journal publication. Below some general and specific comments.

General comments:

In particular checks the journal’s guidelines concerning the reference section. All the reference style is wrong, please check the entire manuscript.

Not all equation style probably was not fixed.

The workflow should correspond to a figure in which the main steps of your work are summarized, (i.e. the data, the processing and analysis steps, and finally the results of your work). Table 1 simply lists the models used, without providing any details of the complete overview of your work.

Conclusions: I think this section must be improved. Specifically, the conclusions should describe in a few sentences what was achieved in the work and not to further present results (i.e. Table 5). Please revise this section.

Specific comments:

Line 34-35: references must be only included in square brackets according to the journal's guidelines.

Line 43: references must be only included in square brackets according to the journal's guidelines.

Line 77-79: references must be only included in square brackets according to the journal's guidelines.

Line 95: references must be only included in square brackets according to the journal's guidelines.

Table 1: check the journal’s guidelines.

Table 5: check the journal’s guidelines.

Author Response

First, the authors would like to thank the reviewer for his valuable comments. The authors agree with reviewer´s #2 minor suggestions and all the points have been addressed. A point-by-point response is provided below:

General comments:

  • In particular checks the journal’s guidelines concerning the reference section. All the reference style is wrong, please check the entire manuscript.

Suggested changes are included in revised manuscript and bibliographical references are placed in square brackets.

  • Not all equation style probably was not fixed.

Equations have been revised.

  • The workflow should correspond to a figure in which the main steps of your work are summarized, (i.e. the data, the processing and analysis steps, and finally the results of your work). Table 1 simply lists the models used, without providing any details of the complete overview of your work.

As suggested, a paragraph explaining a workflow has been added.

  • Conclusions: I think this section must be improved. Specifically, the conclusions should describe in a few sentences what was achieved in the work and not to further present results (i.e. Table 5). Please revise this section.

As suggested, this section has been improved.

Specific comments:

  • Line 34-35, 43,77-79,95 : references must be only included in square brackets according to the journal's guidelines.

Solved

  • Table 1, Table 5: check the journal’s guidelines.

As suggested, tables have been changed and placed in the main text near the first time they are cited